# A Case of *Tinea Corporis* Caused by *Trichophyton benhamiae* var. *luteum* from a Degu and Evolution of the Pathogen’s Taxonomy

**DOI:** 10.3390/jof9111122

**Published:** 2023-11-19

**Authors:** Hiroshi Tanabe, Noriyuki Abe, Kazushi Anzawa

**Affiliations:** 1Department of Dermatology, Tenri Hospital, Tenri 632-8552, Japan; 2Department of Laboratory Medicine, Tenri Hospital, Tenri 632-8552, Japan; abepenem@tenriyorozu.jp; 3Department of Dermatology, Kanazawa Medical University, Uchinada 920-0293, Japan; anzka@kanazawa-med.ac.jp

**Keywords:** *Trichophyton benhamiae* var. *luteum*, *tinea corporis*, degus (*Octodon degus*), terbinafine, Americano-European race (−)

## Abstract

Background: *Trichophyton benhamiae*, an anthropophilic dermatophyte, can cause dermatophytosis in humans and animals with rising zoonotic infections through pets, primarily in Europe. Dermatophytosis from *T. benhamiae* is often misdiagnosed due to its inflammatory symptoms. We report a case of *tinea corporis* from *T. benhamiae* var. *luteum* in a Japanese woman, contracted from pet Czech degus. Case: The 40-year-old patient developed neck papules resembling acne. Initial treatment with topical antibiotics and steroids exacerbated the rash. Fungal elements were not detected by direct potassium hydroxide examination. Skin biopsy confirmed fungal elements in the *stratum corneum* and hair follicles, and *tinea corporis* was diagnosed. Oral terbinafine 125 mg was initiated without topical agents. Erythematous papules appeared on her limbs, determined as a trichophytid reaction. After two months, her skin improved significantly. Fungal culture identified *T. benhamiae* var. *luteum* colonies with a yellowish hue. Mating tests classified the strain as Americano-European race (−) with *MAT1-1* genotype. This was diagnosed as *tinea corporis* from *T. benhamiae* var. *luteum*, likely transmitted from pet Czech degus. Conclusions: The incidence rate of pet-transmitted cutaneous fungal infections may increase in Japan with the trend to keep exotic pets. Dermatologists must recognize dermatophytosis clinical features from anthropophilic dermatophytes to prevent misdiagnosis and understand evolving nomenclature and pathogenesis.

## 1. Introduction

*Trichophyton benhamiae* is a zoophilic dermatophyte. In Europe, it has been increasingly reported in the fields of veterinary medicine and dermatology as a source of infection associated with small animals like rodents, including mice and guinea pigs, causing zoonotic dermatophytosis in humans and animals. Unlike typical anthropophilic dermatophytes such as *Trichophyton rubrum* and *T. tonsurans*, zoophilic and geophilic dermatophytes possess the ability to undergo sexual reproduction, resulting in genetic diversity through mating. However, the nomenclature of these fungi has undergone multiple revisions over time, evolving in the context of mycological studies and technological advancements. Taxonomy often remains a less-explored domain among dermatologists with an interest in diagnosis and treatment, and the shifting nomenclature of fungal species can sometimes serve as a barrier to their engagement in the field. Nevertheless, clinical dermatology must align with the evolving trends in biology.

This case report holds significance for two main reasons. Firstly, due to the recent surge in pet ownership, particularly of animals imported from Europe, it is crucial to raise awareness about the potential spread of *T. benhamiae* var. *luteum* as a causative agent of zoonotic infections in Japan. This case report represents the fourth reported instance of human tinea corporis caused by the yellow phenotype of *T. benhamiae*, classified as Group II by Symoens, in Japan. The misdiagnosis of this condition as contact dermatitis in this case underscores the need for caution among clinical dermatologists.

Secondly, the pathogen has undergone taxonomic changes since it was classified as *Trichophyton benhamiae* in 2017 [1], with the recognition of the subspecies *T. benhamiae* var. *luteum* in 2020 based on morphological characteristics, notably yellow colony formation [2,3]. While these developments are rooted in mycological taxonomy, they may have far-reaching implications, particularly in light of the recent pet boom and the potential emergence of zoonotic dermatophytosis caused by zoophilic fungi. Dermatologists need to comprehend these taxonomic shifts and changing nomenclature, as summarized in this report.

The taxonomy of this fungus may continue to evolve with technological advancements, and insights into the pathogenesis of dermatophytosis caused by this fungus are expected to expand and evolve. The accumulation of various mycological test results from a single case can be valuable for analyzing the pathogenesis and treatment approaches for similar cases in the future. Furthermore, clinical dermatologists should maintain an interest in mycological taxonomy and vigilantly monitor its evolution from a clinical perspective.

## 2. Case Presentation

A 40-year-old Japanese woman presented to our hospital with acne-like papules on the nape of her neck that had enlarged over the course of one month. Her previous doctor had prescribed a combination steroid ointment containing (oxytetracycline 30 mg/g and 10 mg hydrocortisone alcohol/g). The patient had received treatment for *tinea corporis* with topical antifungals one year previously. She had underlying medical conditions, including ovarian cysts and Meniere’s disease. There were no martial arts practitioners in her family, which could have been a possible route of infection.

Clinical examination revealed a well-circumscribed circular erythema with scales, measuring 25 mm × 30 mm in diameter, on the patient’s nape (Figure 1). She did not exhibit any systemic symptoms such as lymphadenopathy or fever, and her blood test results were within the normal range. Due to her prior use of topical antifungal agents, potassium hydroxide (KOH) direct microscopy of the affected area did not reveal any fungal elements. However, a skin biopsy showed significant inflammatory changes in the epidermis and no evidence of follicle destruction in the dermis. Periodic acid-Schiff (PAS) staining revealed the presence of hyphae in the stratum corneum (Figure 2a) and within hair follicles (Figure 2b). Based on these pathological findings, she was diagnosed with *tinea corporis*.

The patient had kept three pet degus for several years (Figure 3). These degus exhibited skin lesions, including hair loss. The patient reported that the degus had been imported from the Czech Republic. However, fungal cultures of tail hairs did not yield any fungal colonies.

## 3. Mycological Findings

Nuchal scales were used as specimens for fungal culture on Sabouraud dextrose agar (SDA) at 27 °C. Within a few days, white colonies with radial expansion were observed. The colony surface gradually developed a fluffy, white, and villus-like appearance with the formation of wrinkles. After two weeks, there was a slight yellowish pigmentation on both sides of the colonies (Figure 4). Similarly, a yellowish pigmentation was observed on both sides of potato dextrose agar (PDA) after two weeks of incubation (Figure 5).

The isolated strain did not form conidia on potato dextrose agar (PDA) or SDA. On the malt extract agar (MEA) medium, it produced sesame-granular microconidia without a spiral body, which stained blue with lactophenol cotton blue staining (Figure 6).

### 3.1. Mating Test

The isolated strain (KMU9518) successfully mated with the Americano-European race RV26678(+) on a 1/10 diluted SDA plate (Figure 7a) and was determined to be the Americano-European race (−). The gymnothecia showed approximately 250 pale yellow spheres, each approximately 350 µm in size (Figure 7b), and several dark yellow ascospores were confirmed when a part of the gymnothecia was cracked open (Figure 8).

### 3.2. Molecular Identification

Kanazawa Medical University Dermatology Department conducted molecular biological identification of isolate KMU9518 and determined the base sequence of the ITS (internal transcribed spacer) region of the ribosomal RNA gene. The sequence was found to be 100% identical to that of the *Trichophyton benhamiae* type strain CBS 623.66, accession number_103705. Furthermore, the same sequence was also identified in the parent strain of the Americano-European race tester strain of *T. benhamiae*.

Regarding the mating type gene involved in mating, PCR was performed following the method described by Symoens et al. (2013) [4]]. Amplification was observed in the PCR with MF3 and MF4 primers, while no amplification was observed with MF1 and MF2 primers. This confirmed that KMU9518 possesses the *MAT1-1* mating type.

### 3.3. Mass Spectrometry by MALDI-TOF MS

After two days of culturing in a liquid medium, the mass spectrum of the causative fungus was analyzed using the MALDI Biotyper™ smart instrument (Bruker Daltonik GmbH, Bremen, Germany). Matching by the exclusive software MALDI Biotyper 3.1 identified *T. benhamiae* with a score value of 1.946.

The mass spectral pattern of *T. benhamiae* var. *luteum*, as described in the literature [3], shows typical wavelengths of 4112, 4680, 6515, and 6530 *m*/*z*. However, in KMU9518, peaks were observed at 4110 and 2081 *m*/*z* in the mass spectrum, with none near 4680 *m*/*z*. Furthermore, the pattern in the visible range was entirely different, and the characteristic waveform differed from that in the literature, which outlines a specific mass spectrum pattern of MALDI-TOF MS for the fungus species. However, our dedicated software analysis and visual observation yielded different results (Figure 9).

The identification of dermatophyte species with diverse morphological characteristics remains challenging when relying solely on mass spectrometry, such as MALDI-TOF MS. Currently, it is considered a complementary method to be used alongside other approaches. The analysis by MALDI-TOF MS depends on known strain databases, which are not yet comprehensive for individual species of dermatophytes, including *Trichophyton benhamiae*. Therefore, for the identification of *T. benhamiae* species, MALDI-TOF MS alone is currently insufficient, and it should be considered as a complementary method alongside other approaches.

Morphological, molecular biological, and mass spectrometry characteristics comprehensively identified the causative fungus in our case as *T. benhamiae*, characterized by a yellow phenotype [3]. Based on these findings, this case was diagnosed as *tinea corporis* caused by *T. benhamiae* var. *luteum*.

### 3.4. Treatment

At the initial consultation, the patient was prescribed oral tetracycline antibiotics and topical application of sodium fusidate. However, there was no improvement in the erythematous rash on the neck. A definitive diagnosis of *tinea corporis* was confirmed through histopathological examination results one week later, which revealed fungal infection within the hair follicles. Consequently, the patient was instructed to discontinue the use of topical antifungal agents, and treatment was initiated with oral terbinafine hydrochloride tablets at a standard dose of 125 mg/day for adults in Japan.

Following this change in treatment, the patient developed multiple pruritic erythematous papules on the limbs and trunk, which were suggestive of a drug eruption (Figure 10). Despite this suspicion, it was determined to be a trichophytid reaction, and treatment with oral terbinafine hydrochloride was continued. The systemic rash showed improvement within several days, and the circular erythematous lesion on the neck improved, ultimately leading to the discontinuation of oral terbinafine hydrochloride after two months (Figure 11). Throughout the treatment period, no abnormalities were observed in peripheral blood cell counts or biochemical tests.

## 4. Discussion

### 4.1. Evolution of the Taxonomy of Trichophyton benhamiae

*Trichophyton benhamiae* was initially described as the anamorphic stage within the fungus known as *Arthroderma benhamiae*, which is capable of sexual reproduction within the *Trichophyton mentagrophytes* (anamorph) species complex (Ajello and Cheng 1967) [5].

Mating experiments were conducted using 21 preserved strains of *Trichophyton mentagrophytes* var. *granulosum* (Neveu-Lemaire 1911). Fertile cleistothecia containing ascospores resulted from the cross between strain TM20 from a human in Missouri and strain TM17 from a dog in Illinois. This hybrid was named after mycologist Dr. Rhoda Benham for guinea pig transplant experiments. The type strain, NCDC B765d, was designated. For *A. benhamiae*, representing the sexual stage, two strains with both mating types were registered as live cultures: CDC X-797A:ATCC16781 = CBS 623.66 = type strain of mt+ [TM20 × TM17 = ATCC 16781] and CDC X-798a:ATCC16782 = CBS 624.66 = type strain of mt− [TM20 × TM17 = ATCC 16782].

Takashio (1974, 1977) later classified two groups based on biological compatibility experiments, distinguishing between Americano-European and African races [6]. Tester strains for the Americano-European race included IHEM 3287 = RV 26678 (SA-3 from Takashio 1974) mt+ and IHEM 3288 = RV 26680 (SA-5 from Takashio 1974) mt−. Additionally, by crossing strains RV27926(+) from humans in South Africa and strain RV25293(−) from dogs in Mozambique, ascospores were obtained, and the resulting F1 strains RV30000(+) and RV30001(−) were designated as reference strains for the African race.

At the 18th International Botanical Congress in 2011, the principle ‘one fungus, one name’ was established. This principle aims to bring consistency and clarity to the naming of fungi. Accordingly, proliferative forms through asexual reproduction are termed ‘anamorphs’, while those through sexual reproduction are termed ‘teleomorphs’. Both of these forms, along with their respective names, are collectively known as ‘holomorphs’. This approach seeks to provide a unified name for each fungal species, eliminating confusion that may arise from having different names for different stages of fungal development [7].

In 2008, Kano conducted research on the *MAT* genes associated with fungal mating ability, identifying and analyzing *MAT* genes [8]. In 2012, based on this work, a new classification method for dermatophytes in the perfect state was proposed [9].

In 2013, Symoens et al. differentiated the yellow phenotype, which forms characteristic yellow colonies among the Americano-European race, from the white phenotype, both morphologically and molecularly [4].

In 2017, de Hoog et al. [1] reevaluated this taxonomic arrangement through molecular methods, unifying the species name for the teleomorph *Arthroderma benhamiae* as *Trichophyton benhamiae*. They further classified *Trichophyton* species into *T. mentagrophytes*-series, *T. benhamiae*-series, *T. bullosum*, and *T. rubrum* complex. The *Trichophyton benhamiae* complex included six species: *T. benhamiae*, *T. bullosum*, *T. erinacei*, *T. eriotrephon*, *T. concentricum*, and *T. verrucosum*.

In 2020, Čmoková et al. thoroughly investigated *T. benhamiae* and related strains using a polyphasic approach and proposed detailed species descriptions. They introduced *T. japonicum* and *T. europaeum* as new species alongside *T. benhamiae* within the white phenotype. For the yellow phenotype, due to minor differences in the type strain of *T. benhamiae* and its sequences, they designated a new variety, *T. benhamiae* var. *luteum*, signifying the yellow phenotype [3].

Although taxonomy is challenging for clinicians, its evolution is very interesting as a way to understand human infections caused by microorganisms with both complete and incomplete life cycles. The changes in dermatophyte taxonomy and alterations in the nomenclature of fungal species should continue to be closely monitored from the perspective of clinical dermatology in the future. Figure 12 provides a visual representation of the taxonomic changes discussed in this paper.

### 4.2. Epidemiology and Clinical Impact of T. benhamiae var. luteum Infections in Japan

The incidence rate of *T. benhamiae* infections in humans has been increasing worldwide over the last 15 years, especially in infants in Germany [10]. *T. benhamiae* infections in humans often manifest with extensive inflammation and severe skin lesions, such as *tinea corporis*, *tinea cruris*, and kerion celsi, often associated with secondary bacterial infections [11]. Recently, in Germany, there has been a notable increase in *T. benhamiae* infections, particularly among individuals who have been spending more time at home with their pets, especially since the onset of the COVID-19 pandemic [12].

In Japan, as late as 1980, *Arthroderma benhamiae,* the original name for *T. benhamiae*, had not been confirmed [13,14]. Subsequently, in 2000, Kawasaki et al. reported the first case in Japan of human *tinea corporis* caused by *A. benhamiae* from a pet rabbit [15]. Since then, there have been sporadic cases of such infections accompanying the increase in pet ownership.

We conducted a literature review of clinical cases of *tinea* caused by *A. benhamiae* (limited to genotypes I and II, excluding Type III and *T. erinacei*) in Japan. We identified 26 cases, including our own, reported since 2000 [15,16,17,18,19,20,21,22,23,24,25,26,27,28,29,30,31,32,33]. Among these cases, 21 were female, and five were male [16,19,24,29]. Nineteen of the 26 cases were aged under 30 years old. There were familial instances, including three parent–child cases [15,16,20] and two sibling cases [26,28]. Of the 26 cases, 20 individuals owned pets. Among these pet owners, 10 had rabbits, six had guinea pigs, and four had degus [27,29]. Of the six that did not own pets, four had a history of contact with animals, including two that worked at pet shops [23,32], and two that had contact with small animals, such as rabbits in a zoo [25,30]. Regarding the remaining two individuals, one had no recorded history of contact with animals [31], while the other was a mycologist [21]. 

Reports of the yellow phenotype of *T. benhamiae* (referred to as Group II by Symoens) in Japan, including this case, amount to four cases. The cases reported by Mochizuki, Kobayashi, and others involved two sisters, aged 27 and 25, who kept guinea pigs as pets, and the isolated strains were registered as KMU6909 and KMU7000 [26]. The case reported by Yanagihara involved a 22-year-old woman who kept pets such as degus and guinea pigs, and the strain was identified as KMU10220 [27]. These three cases were reported prior to the classification by Čmoková in 2020. Subsequently, they were reevaluated through molecular biology techniques and identified as Group II. In 2014, Hiruma et al. detected strain of *Arthroderma benhamiae* Group II in Japan, originated from a degu, and was registered as NUBS13001; DDBJ accession number: AB973435 [34]. Clinical reports of *T. benhamiae* var. *luteum* in Japan have included only our case. Čmoková has documented three human-derived strains of *T. japonicum* detected in Japan in reference [3]. These include VUT 00003-2 [35], NUBS 12001, and NUBS 13002 = CCF 6488 [34]. However, since there is no detailed clinical information available for these three strains in the referenced literature, they have not been included in the aforementioned data compilation. Additionally, there has been no clinical report of *T. japonicum* infected humans in Japan.

## 5. Conclusions

The case reported in this study was a patient who kept Czech degus as pet. She was infected with *T. benhamiae* and it is possible that *T. benhamiae* was introduced and spread in Japan through the transportation and commercialization of animals. This background includes issues such as the breeding environment and living conditions of pet animals, irresponsible pet owners, and the trend of staying at home due to COVID-19 infection. Although not yet a significant problem in Japan, the incidence rate of dermatophytosis caused by anthropophilic fungi may increase in the future, as seen in Germany since 2020 [12]. These diseases can be challenging for dermatologists to diagnose correctly, highlighting the importance of understanding the changing nomenclature of these animal-derived dermatophytes, their clinical manifestations, and appropriate management approaches [12,13]. Dermatologists should remain vigilant to avoid misdiagnosis, as these diseases can be challenging to identify correctly.

## Figures and Tables

**Figure 1 jof-09-01122-f001:**
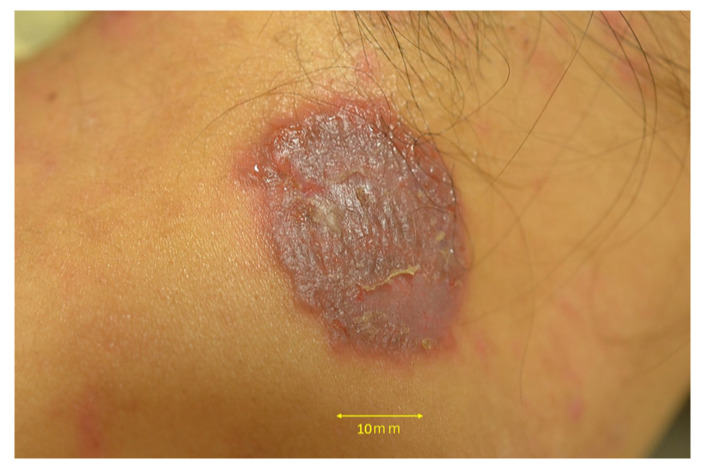
Scaly erythema lesion 25 mm × 30 mm in diameter, on the patient’s nape.

**Figure 2 jof-09-01122-f002:**
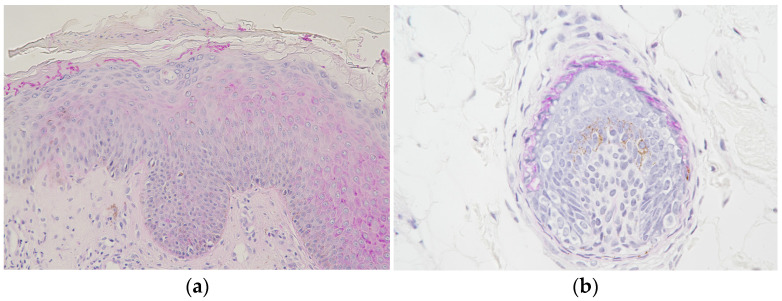
(**a**). PAS stain of skin biopsy revealing hyphae in the *stratum corneum*. Epidermal acanthosis is present, and the dermis shows infiltration of inflammatory cells around blood vessels. 100× magnification. (**b**). PAS stain. Enlarged view of the subcutaneous hair bulb. Numerous hyphae stained strongly were observed in the inner root sheath. 200× magnification.

**Figure 3 jof-09-01122-f003:**
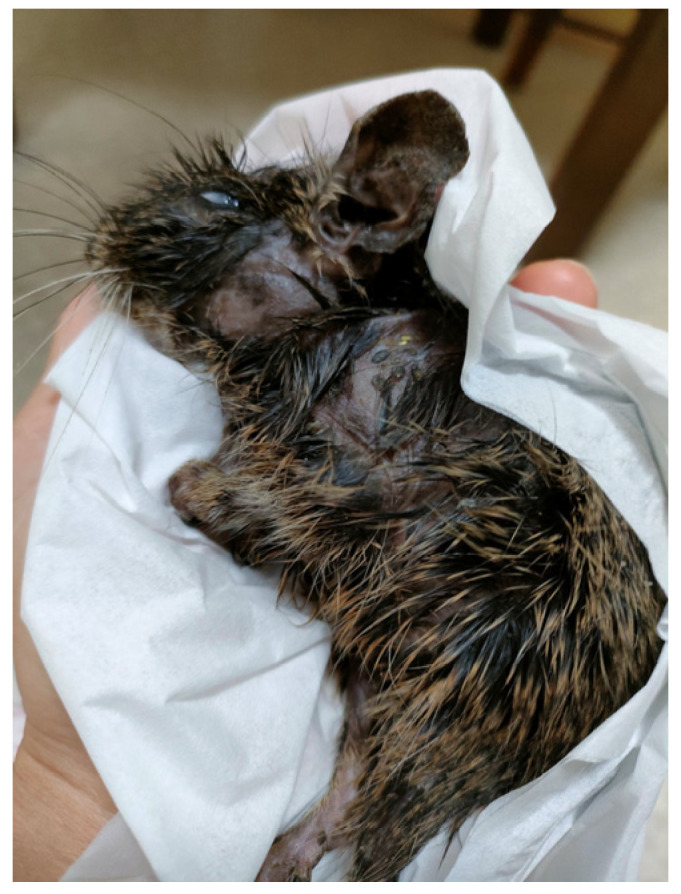
A Czech degu, kept as a pet by the patient. Hair loss progressively extended from the cheeks to the entire body, becoming refractory and emitting a foul odor, eventually leading to demise of the degu five years later.

**Figure 4 jof-09-01122-f004:**
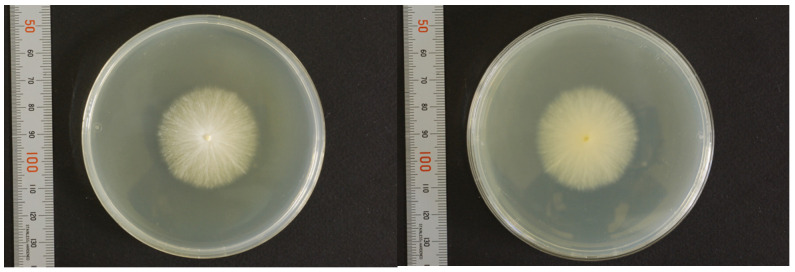
Fungal culture findings in affected scales (SDA, 27 °C, 2-weeks).

**Figure 5 jof-09-01122-f005:**
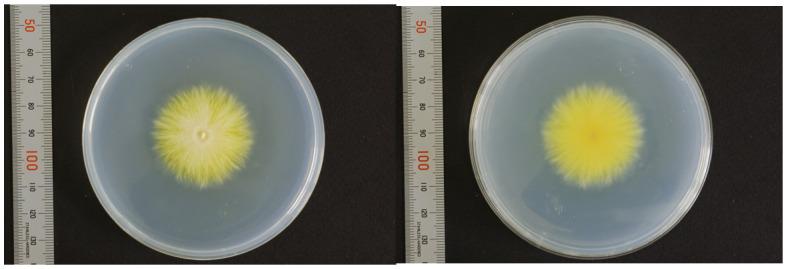
Fungal culture findings in affected scales (PDA, 27 °C, 2-weeks). Both sides of the colonies turned yellowish.

**Figure 6 jof-09-01122-f006:**
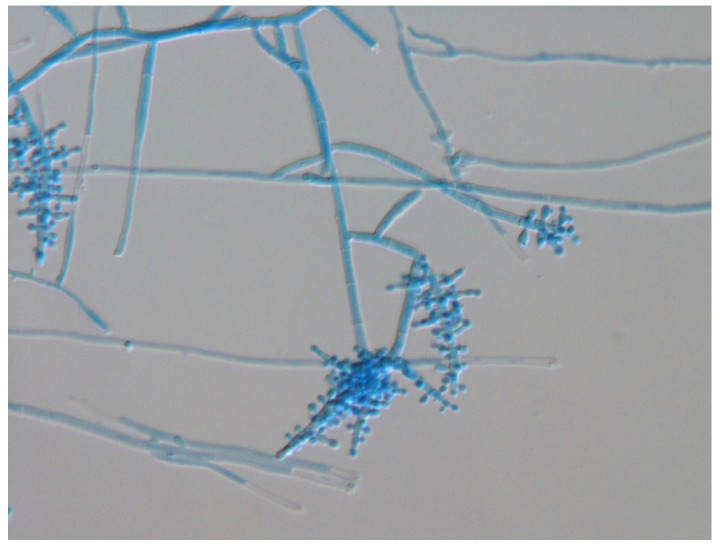
Microscopic examination of lactophenol cotton blue stained samples of slide cultures revealed sesame-shaped microconidia without a spiral body. (MEA (OxoidTM) at 27 °C after 2 weeks). 400× magnification.

**Figure 7 jof-09-01122-f007:**
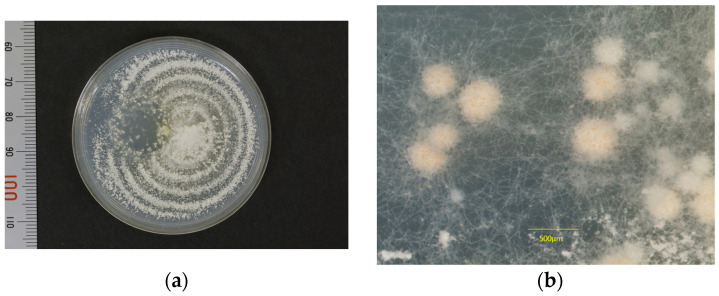
(**a**) Mating test results: In the photograph depicting the overall view after 3 months on a 1/10 diluted SDA plate, one can see the Americano-European race RV26678(+) (**center-right**) and the case strain (KMU9518) (**center-left**). In the central region arranged in a circular pattern, we observed the formation of exactly 250 pale yellowish spherical gymnothecia of varying sizes. (**b**) In the enlarged view of the central yellowish gymnothecia shown in (**a**), the diameter measures approximately 400 µm. Their sizes were measured using an SM2 eyepiece scale, ranging from 250 to 400 µm in diameter (min 250, max 450), with an average of 350 µm, based on measurements of 41 gymnothecia.

**Figure 8 jof-09-01122-f008:**
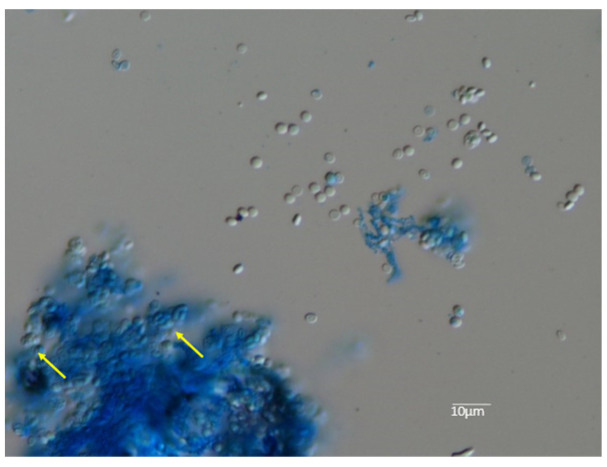
This is an enlarged view of asci (indicated by the yellow arrow at the bottom left) and their contents, which are ascospores (located to the right in the photo). When one of the yellowish gymnothecia was cracked open, disc-shaped ascospores were detected scattered. The diameter of ascospores ranges from 3 to 5 µm, and staining was performed using lactophenol cotton blue.

**Figure 9 jof-09-01122-f009:**
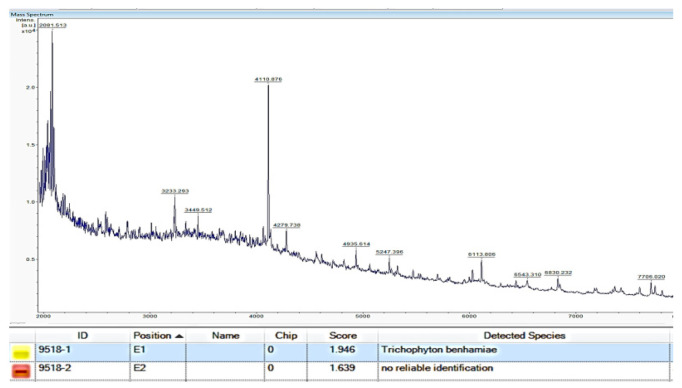
The mass spectral pattern of strain (KMU9518). Log score was 1.946, and *T. benhamiae* was detected below the cutoff value 2.0 for “genus identification” as recommended by the manufacturer.

**Figure 10 jof-09-01122-f010:**
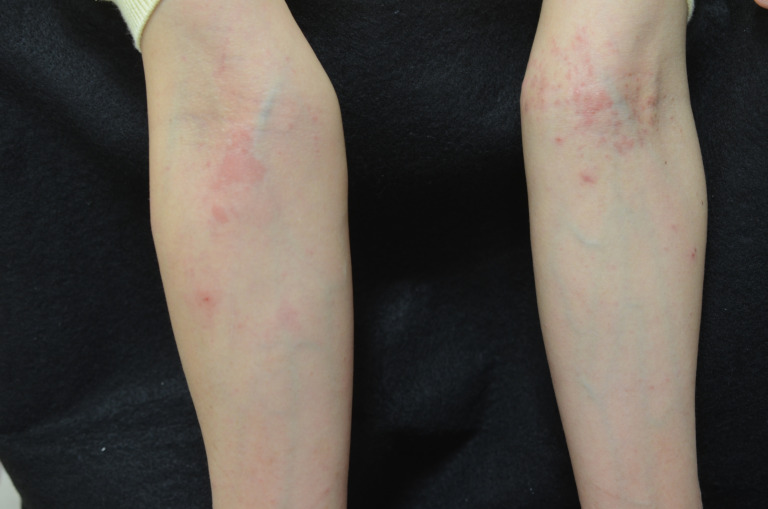
A symmetric pruritic rash emerged throughout the body after the initiation of treatment. KOH direct microscopy examination showed negative results. Suspecting a trichophytid reaction, treatment with oral terbinafine hydrochloride was continued.

**Figure 11 jof-09-01122-f011:**
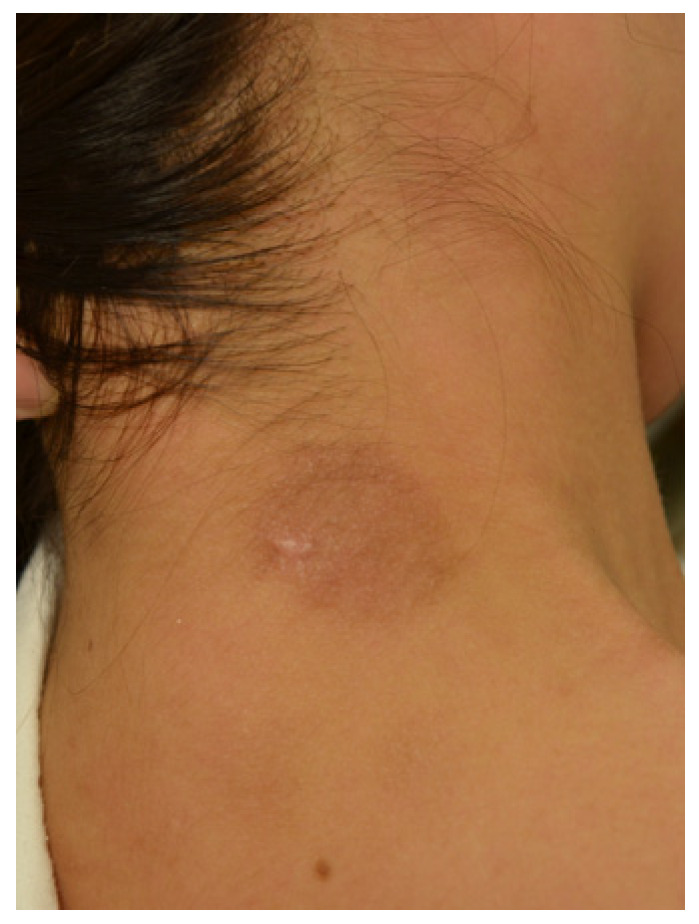
Clinical presentation of the affected area after two months of treatment. Scars from the biopsy site remained, but the erythematous lesions resolved with hyperpigmentation and healing.

**Figure 12 jof-09-01122-f012:**
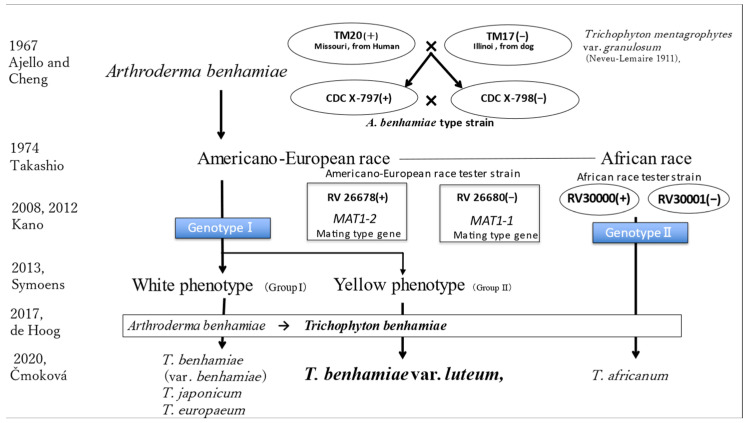
Taxonomic evolution of *Trichophyton benhamiae.* The figure illustrates the taxonomic evolution of *Trichophyton benhamiae*, a dermatophyte species. It outlines key milestones in the species’ classification history, including the recognition of mating types, the proposal of new classification methods, and the differentiation of phenotypic variations within the species. The figure helps to trace changes in the nomenclature and taxonomy of *T. benhamiae* over time, highlighting its significance in the field of mycology and dermatology [1,3,4,5,6,8,9].

## Data Availability

Data are contained within the article.

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
