# Peer review of "A Case of *Tinea Corporis* Caused by *Trichophyton benhamiae* var. *luteum* from a Degu and Evolution of the Pathogen’s Taxonomy"

_jof, 2023, doi:10.3390/jof9111122_

Round 1
Reviewer 1 Report
Comments and Suggestions for Authors
This report presents a case of tinea corporis caused by Trichophyton benhamiae var. luteum infected by degu. This report is not interesting for readers of JoF. Clinical lesions and treatment are not different from the classical presentation and treatment of dermatophyte infection and transmission by the degu has not been proved.
Comments on the Quality of English LanguageMust be improved
Author Response
Dear Reviewer 1,
I would like to express my sincere gratitude for your valuable review of our manuscript.
As per your suggestions, we have made comprehensive revisions to clarify the thesis and explicitly state the authors' key points throughout the manuscript. Additionally, we have adjusted the title to align with the revised content. With the assistance of a trusted native English-speaking scientist, we have proofread the manuscript and resubmitted it. We kindly request your consideration for publication in the esteemed Journal of Fungi, hoping that our revisions meet with your approval.
Regarding the nature of this case report, I acknowledge that it presents a rather typical clinical picture and routine laboratory findings of superficial dermatophyte infection, as you pointed out. However, my decision to submit this report to your distinguished journal was driven by my frustration with the misdiagnosis by dermatologists, a prevalent issue in Japan. While Japanese dermatologists display great enthusiasm for cosmetic and immunological aspects, there is a significant lack of interest in common diseases such as dermatophyte infections. Even basic clinical tests like direct microscopy are often overlooked, leading to the misdiagnosis of superficial dermatophyte infections by many dermatologists. In the case we reported, the patient consulted several dermatologists who misdiagnosed the condition as contact dermatitis or bacterial infections before finally seeking our expertise, which resulted in a confirmed diagnosis through pathological examination and subsequent successful treatment. Eager to shed light on the frustrations experienced by misdiagnosed patients suffering from itching and visible skin discomfort, I intentionally chose to submit this report to your globally renowned journal.
Through this case report, I aim to convey two crucial points to clinical dermatologists. Firstly, even in cases with atypical clinical presentations, clinicians should inquire about pet ownership during patient history taking to differentiate dermatophyte infections caused by zoophilic fungi. Secondly, clinicians should not mistake the trichophytid reaction induced by oral terbinafine as a drug allergy when treating dermatophyte infections with dermal inflammation. Clinicians should be aware that transient worsening of skin symptoms is a common occurrence in the treatment of dermal dermatophyte infections.
Lastly, as part of this submission, I have enclosed an English translation of a letter received from the patient when seeking permission for the use of their photographs. The letter from the patient reflects a sense of distrust towards dermatologists' misdiagnoses and underscores the need for awareness. The patient has also given their consent for publication in your esteemed journal. I kindly request you to review it for your reference.
Dr. Hiroshi Tanabe
I am delighted to hear about your manuscript submission. I hope that misdiagnoses leading to worsened symptoms can be avoided in the future, and accurate diagnoses will prevail. I grant permission for the use of the photos I sent via email. I wish you continued success in your work and good health.
Y.K.
Having considered the points mentioned above, I humbly request your permission for publication in your prestigious journal.
Sincerely,
Hiroshi Tanabe

Reviewer 2 Report
Comments and Suggestions for Authors
The authors present an interesting case report about a Trichophyton benhamiae var. luteum infection. The typically infection source of this species variant were pets as guinea pigs. In this report, the authors show that also an infected degu transmit the disease. The authors mention the difficulty to obtain the correct diagnosis if some analyses like direct microscopy of scraped skin particles failed. The authors show all available diagnostic tools to identify the correct species inclusive MALDI-TOF analyses and sequencing of ITS region.
Minor points:
The authors should include a size bar in microscopic image in Figure 6 and in Figure 8
Author Response
Dear Reviewer 2,
Response
We thank Reviewer 2 for taking the time to read our article and contribute to its contents. We are please that this Reviewer found it interesting.
I have reviewed and revised the logical structure of the paper, adding content to make the author's arguments more clear. I have also modified the title to make the main point more evident.
- **Size Bar in Microscopic Images**: We acknowledge your suggestion regarding the inclusion of a size bar in the microscopic images in Figure 7a, 7b, and Figure 8. Additionally, we have added magnification details to Fig 6. We have now revised these figures to include a size bar for clarity and improved understanding."
Once again, we would like to express our gratitude for your valuable feedback, which has undoubtedly enhanced the quality of our manuscript. We believe that these revisions have strengthened our work, and we hope that it now meets the standards for publication in Journal of Fungi.
Sincerely,
Hiroshi Tanabe
[Title] jof-2582074 " A case of tinea corporis caused by Trichophyton benhamiae var. luteum from a degu and evolution of the pathogen’s taxonomy"
Reviewer 3 Report
Comments and Suggestions for Authors
The manuscript jof-2582074 entitled “A case of tinea corporis caused by Trichophyton benhamiae var. luteum infected by degu” has been reviewed.
The significance of this report is the emergence of T. benhamiae var. luteum in Asia and its possible importation route through the infected / carrier of pet degu from Czech, an endemic area of this fungus. Comments are listed here,
1. Extensive scientific English editing is suggested to correct grammar errors and tweak some sentences to make the flow of the manuscript better.
2. L58, 62: abbreviation in the parentheses, e.g. Sabouraud dextrose agar (SDA)
3. L77 The isolate KMU 9518 was confirmed to be AE race, MT(-). Is this designation equal to MAT1-1-1 or MAT1-2-1 per publication of Čmoková et al. (ref. 3)? Please identify the MAT type of your isolate.
4. L83: What does “3M” mean?
5. Figure 8. Please replace it with high-power figures of the ascocarp (gymnothcia?) and asci/ascospores along with a scale bar.
6. L95, L103: bacterium -->fungus?
7. L117 Why start from a low dose of terbinafine? Any concerns or considerations?
8. L123 depigmentation-->do you mean subsiding of the erythema?
9. L126 What is the reason for measuring β-D-glucan for a superficial dermatophyte infection?
10. L136-138 Please rewrite this sentence because Trichophyton benhamiae was a name for anamorph (imperfect state).
11. L140 “perithecia”—it’s “cleistothecia” in the original publication
12. L156-159 consider deleting this paragraph or rewriting it because the text itself doesn’t explain the principle of “one fungus, one name”
13. L165-168: T. benhamiae-series and T. benhamiae complex-->which one is correct?
14. L173 subspecies and variety are different taxonomic ranks; var. is the abbreviation of variety.
15. L180-180. This a figure.
16. L195-202 Please add all references for reported cases in Japan. Are these isolates now re-identified as T. japonicum? Because this fungus is the most prevalent species in Japan based on the current taxonomy of T. benhamiae species complex.
17. The review of taxonomic change of the Trichophyton benhamiae species complex in Discussion is interesting but the content is better to be more concise and easier to understand.
Comments on the Quality of English LanguageExtensive scientific English editing is suggested to correct grammar errors and tweak some sentences to make the flow of the manuscript better.
Author Response
Dear Reviewer 3,
I would like to express my sincere gratitude for your valuable review of our manuscript. Your detailed and expert insights, along with your comments, have provided us with numerous learning opportunities during the process of revising the manuscript. We truly appreciate your precise and insightful feedback.
As per your suggestions, we have made comprehensive revisions throughout the manuscript to clarify the thesis and explicitly state the authors' key points. Furthermore, in alignment with the revised content, we have updated the title to "A case of tinea corporis caused by Trichophyton benhamiae var. luteum from a degu and evolution of the pathogen’s taxonomy." With the assistance of a trusted native English-speaking scientist, we have proofread the manuscript and resubmitted it. The items we have modified in accordance with your comments are outlined in the attached file. We kindly request your reevaluation and hope that you would consider our manuscript for publication in the esteemed Journal of Fungi.
Sincerely,
Hiroshi Tanabe

Round 2
Reviewer 3 Report
Comments and Suggestions for Authors
L60 Suggest deleting the word “teleomorphic”
L141 No_103705àNR_103705
L201 Trichophyton mentagrophytesàitalic
L252 A figure or a table? Inconsistency in caption and legend.
L292-293 In the original description of Trichophyton japonicum by Čmoková et al. (reference 3 of this manuscript), the author stated that the epithet of this species comes from “the origin of the majority of the examined strains”. Of the examined materials, some strains are from human living in Japan, including “Saitama, human, 2000 (VUT 00003–2), human, 2013 (NUBS 12001), and human, unknown (NUBS 13002=CCF 6488)”. So, there should be human case in Japan at least since 2000. Please review if these cases have been reported in any form or language and make a amendment.
Author Response
Dear Reviewer 3,
We appreciate your thoughtful review and your valuable comments on our manuscript. Your feedback has been instrumental in improving the quality and accuracy of our work. Here are our responses to your points:
- L60: We will delete the word "teleomorphic" as suggested.
- L141: We will correct "No_103705" to "NR_103705."
- L201: "Trichophyton mentagrophytes" will be formatted in italics as recommended.
- L252: We apologize for any confusion. The element in question is not a figure but a table. We have updated the caption to reflect this change. The revised caption is as follows:
**Table 1. Taxonomic Evolution of Trichophyton benhamiae.**
*This table illustrates the taxonomic evolution of Trichophyton benhamiae, a dermatophyte species. It outlines key milestones in the species' classification history, including the recognition of mating types, the proposal of new classification methods, and the differentiation of phenotypic variations within the species. The table helps to trace the changes in the nomenclature and taxonomy of T. benhamiae over time, highlighting its significance in the field of mycology and dermatology.*
- L292-293: We would like to express our gratitude for your diligent review of our manuscript. We have thoroughly investigated the issue regarding Trichophyton japonicum, specifically its origin from human sources, as referenced in your comments on citation 3. Below, we provide a detailed response along with the relevant references.
Regarding the entry "Saitama, human, 2000 (VUT 00003–2), human," this information is sourced from the publication by Saito K in J. Vet. Med. Sci. 63(8): 929-931, 2001. Please refer to the attached file for the references in PDF format.
It involves a case report in veterinary medicine, where the strain VUT 00003 was isolated from a 4-month-old female dwarf rabbit with alopecia. The report mentions that multiple owners of the rabbit developed rashes on their limbs, face, and axillary regions. However, specific details about the age of the owners and their skin conditions are not provided. The strain isolated from the owners is VUT 00003-2.
In a publication by Kano R, Medical Mycology, November 2008, 46, 739-744, the strains A. benhamiae: mating type(-) VUT-00003 Rabbit (isolated at Saitama) and A. benhamiae (-) VUT-00003-2 Human (isolated at Saitama) are mentioned. It is worth noting that "VUT" refers to the Veterinary Medicine University of Tokyo, Japan's library.
Concerning "2013 (NUBS 12001), and human, unknown (NUBS 13002=CCF 6488)," this information is sourced from the publication by Hiruma J in Mycopathologia (2015) 179:219-223. Please refer to reference (34) in the paper for this information. In Table 1 of this publication, the strains NUBS 12001 and NUBS 13002 are listed, but detailed information about the patients or clinical presentations is not provided. Additionally, there is no registration of accession numbers, and no matching cases were found in the existing literature. In the phylogenetic tree based on the ITS region (Figure 2) in the same paper, NUBS 12001 and 13002 both belong to Group I and are recorded as the GenBank accession number of ITS region analysis of dermatophytes AB048192 (= KMU4136).
Regarding these strains, we contacted Dr. Kano R directly and received the following response: "These strains were transferred from Dr. Kano to Dr. Vit in the Czech Republic in response to Dr. Vit's request for Japanese A. benhamiae strains. Subsequently, they were named T. japonicum."
Furthermore, NUBS refers to Nihon University College of Bioresource Sciences, and CCF is likely associated with Charles University in the Czech Republic. We recommend contacting them directly for further details.: [https://www.natur.cuni.cz/biology/botany/structure/culture-collection-of-fungi-ccf](https://www.natur.cuni.cz/biology/botany/structure/culture-collection-of-fungi-ccf)
Based on our investigation, we were unable to find detailed case reports of Trichophyton japonicum infections in humans in Japan, at least since 2000.
We would like to express our sincere gratitude for your dedication to this review process and your invaluable contributions to the advancement of our research. Your thoughtful insights have significantly enhanced the quality of our manuscript, and we deeply appreciate your expertise and time.
Sincerely,
Hiroshi Tanabe

Round 3
Reviewer 3 Report
Comments and Suggestions for Authors
All questions have been properly answered.
As you mentioned in the response, if you accept the species concept of the Čmokova and Hubka, these strains, either from animals or humans, are now T. japonicum rather than A. benhamiae regardless of their name in the original publication.